# Fine-grained Entity Recognition with Reduced False Negatives and Large Type Coverage

**Abhishek**                                                    abhishek.abhishek@iitg.ac.in
*Indian Institute of Technology Guwahati,*
*Guwahati, Assam, India*

**Sanya Bathla Taneja**[*]                                      sanyabt11@gmail.com
**Garima Malik**[*]                                             annu.2353@gmail.com
*Indira Gandhi Delhi Technical University for Women,*
*Kashmere Gate, Delhi, India*

**Ashish Anand**                                                anand.ashish@iitg.ac.in
**Amit Awekar**                                                 awekar@iitg.ac.in
*Indian Institute of Technology Guwahati,*
*Guwahati, Assam, India*

## Abstract

Fine-grained Entity Recognition (FgER) is the task of detecting and classifying entity mentions to a large set of types spanning diverse domains such as biomedical, finance and sports. We observe that when the type set spans several domains, detection of entity mention becomes a limitation for supervised learning models. The primary reason being lack of dataset where entity boundaries are properly annotated while covering a large spectrum of entity types. Our work directly addresses this issue. We propose Heuristics Allied with Distant Supervision (HAnDS) framework to automatically construct a quality dataset suitable for the FgER task. HAnDS framework exploits the high interlink among Wikipedia and Freebase in a pipelined manner, reducing annotation errors introduced by naively using distant supervision approach. Using HAnDS framework, we create two datasets, one suitable for building FgER systems recognizing up to 118 entity types based on the FIGER type hierarchy and another for up to 1115 entity types based on the TypeNet hierarchy. Our extensive empirical experimentation warrants the quality of the generated datasets. Along with this, we also provide a manually annotated dataset for benchmarking FgER systems.[1]

## 1. Introduction

In the literature, the problem of recognizing a handful of coarse-grained types such as *person*, *location* and *organization* has been extensively studied [Nadeau and Sekine, 2007, Marrero et al., 2013]. We term this as Coarse-grained Entity Recognition (CgER) task. For CgER, there exist several datasets, including manually annotated datasets such as CoNLL [Tjong Kim Sang and De Meulder, 2003] and automatically generated datasets such as WP2 [Nothman et al., 2013]. Manually constructing a dataset for FgER task is an expensive and

---

*. The authors contributed to the work during their internship at the Indian Institute of Technology Guwahati.

1. The code and datasets to replicate the experiments are available at https://github.com/abhipec/HAnDS.

time-consuming process as an entity mention could be assigned multiple types from a set of thousands of types.

In recent years, one of the subproblems of FgER, the Fine Entity Categorization or Typing (Fine-ET) problem has received lots of attention particularly in expanding its type coverage from a handful of coarse-grained types to thousands of fine-grained types [Murty et al., 2018, Choi et al., 2018]. The primary driver for this rapid expansion is exploitation of cheap but fairly accurate annotations from Wikipedia and Freebase [Bollacker et al., 2008] via the distant supervision process [Craven and Kumlien, 1999]. The Fine-ET problem assumes that the entity boundaries are provided by an oracle.

We observe that the detection of entity mentions at the granularity of Fine-ET is a bottleneck. The existing FgER systems, such as FIGER [Ling and Weld, 2012], follow a two-step approach in which the first step is to detect entity mentions and the second step is to categorize detected entity mentions. For the entity detection, it is assumed that all the fine-categories are subtypes of the following four categories: *person*, *location*, *organization* and *miscellaneous*. Thus, a model trained on the CoNLL dataset [Tjong Kim Sang and De Meulder, 2003] which is annotated with these types can be used for entity detection. Our analysis indicates that in the context of FgER, this assumption is not a valid assumption. As a face value, the *miscellaneous* type should ideally cover all entity types other than *person*, *location*, and *organization*. However, it only covers 68% of the remaining types of the FIGER hierarchy and 42% of the TypeNet hierarchy. Thus, the models trained using CoNLL data are highly likely to miss a significant portion of entity mentions relevant to automatic knowledge bases construction applications.

Our work bridges this gap between entity detection and Fine-ET. We propose to automatically construct a quality dataset suitable for the FgER, i.e, both Fine-ED and Fine-ET using the proposed HAnDS framework. HAnDS is a three-stage pipelined framework wherein each stage different heuristics are used to combat the errors introduced via naively using distant supervision paradigm, including but not limited to the presence of large false negatives. The heuristics are data-driven and use information provided by hyperlinks, alternate names of entities, and orthographic and morphological features of words.

Using the HAnDS framework and the two popular type hierarchies available for Fine-ET, the FIGER type hierarchy [Ling and Weld, 2012] and TypeNet [Murty et al., 2018], we automatically generated two corpora suitable for the FgER task. The first corpus contains around 38 million entity mentions annotated with 118 entity types. The second corpus contains around 46 million entity mentions annotated with 1115 entity types. Our extensive intrinsic and extrinsic evaluation of the generated datasets warrants its quality. As compared with existing automatically generated datasets, supervised learning models trained on our induced training datasets perform significantly better (approx 20 point improvement on micro-F1 score). Along with the automatically generated dataset, we provide a manually annotated corpora of around thousand sentences annotated with 117 entity types for benchmarking of FgER models. Our contributions are highlighted as follows:

- We analyzed that existing practice of using models trained on CoNLL dataset has poor recall for entity detection in the Fine-ET setting, where the type set spans several diverse domains. (Section 3)

- We propose HAnDS framework, a heuristics allied with the distant supervision approach to automatically construct datasets suitable for FgER problem, i.e., both fine entity detection and fine entity typing. (Section 4)

- We establish the state-of-the-art baselines on our new manually annotated corpus, which covers 2.7 times more finer-entity types than the FIGER gold corpus, the current de facto FgER evaluation corpus. (Section 5)

The rest of the paper is organized as follows. We describe the related work in Section 2, followed by a case study on entity detection problem in the Fine-ET setting, in Section 3. Section 4 describes our proposed HAnDS framework, followed by empirical evaluation of the datasets in Section 5. In Section 6 we conclude our work.

## 2. Related Work

We majorly divide the related work into two parts. First, we describe work related to the automatic dataset construction in the context of the entity recognition task followed by related work on noise reduction techniques in the context of automatic dataset construction task.

In the context of FgER task, [Ling and Weld, 2012] proposed to use distant supervision paradigm [Black et al., 1998] to automatically generate a dataset for the Fine-ET problem, which is a sub-problem of FgER. We term this as a Naive Distant Supervision (NDS) approach. In NDS, the linkage between Wikipedia and Freebase is exploited. If there is a hyperlink in a Wikipedia sentence, and that hyperlink is assigned to an entity present in Freebase, then the hyperlinked text is an entity mention whose types are obtained from Freebase. However, this process can only generate positive annotations, i.e., if an entity mention is not hyperlinked, no types will be assigned to that entity mention. The positive only annotations are suitable for Fine-ET problem but it is not suitable for learning entity detection models as there are large number of false negatives (Section 3). This dataset is publicly available as FIGER dataset, along with a manually annotated evaluation corpra. The NDS approach is also used to generate datasets for some variants of the Fine-ET problem such as the Corpus level Fine-Entity typing [Yaghoobzadeh et al., 2018] and Fine-Entity typing utilizing knowledge base embeddings [Xin et al., 2018]. Much recently, [Choi et al., 2018] generated an entity typing dataset with a very large type set of size 10k using head words as a source of distant supervision as well as using crowdsourcing.

In the context of CgER task, [Nothman et al., 2008, 2009, 2013] proposed an approach to create a training dataset for CgER task using a combination of bootstrapping process and heuristics. The bootstrapping was used to classify a Wikipedia article into five categories, namely *PER*, *LOC*, *ORG*, *MISC* and *NON-ENTITY*. The bootstrapping requires initial manually annotated seed examples for each type, which limits its scalability to thousands of types. The heuristics were used to infer additional links in un-linked text, however the proposed heuristics limit the scope of entity and non-entity mentions. For example, one of the heuristics used mostly restricts entity mentions to have at least one character capitalized. This assumption is not true in the context for FgER where entity mentions are from several diverse domains including biomedical domain.

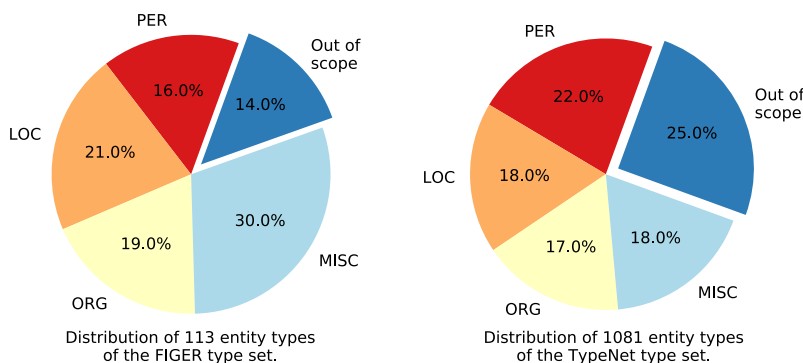

Figure 1: The entity type coverage analysis of the FIGER and the TypeNet type set. This illustrates that a significant portion of entity types are not a descendant of any of the four CoNLL types.

There are other notable work which combines NDS with heuristics for generating entity recognition training dataset, such as [Al-Rfou et al., 2014] and [Ghaddar and Langlais, 2017]. However, their scope is limited to the application of CgER. Our work revisits the idea of automatic corpus construction in the context of FgER. In HAnDS framework, our main contribution is to design data-driven heuristics which are generic enough to work for over thousands of diverse entity types while maintaining a good annotation quality.

An automatic dataset construction process involving heuristics and distant supervision will inevitably introduce noise and its characteristics depend on the dataset construction task. In the context of the Fine-ET task [Ren et al., 2016b, Gillick et al., 2014], the dominant noise in false positives. Whereas, for the relation extraction task both false negatives and false positives noise is present [Roth et al., 2013, Phi et al., 2018].

## 3. Case study: Entity Detection in the Fine Entity Typing Setting

In this section, we systematically analyzed existing entity detection systems in the setting of Fine-Entity Typing. Our aim is to answer the following question : How good are entity detection systems when it comes to detecting entity mentions belonging to a large set of diverse types? We performed two analysis. The first analysis is about the type coverage of entity detection systems and the second analysis is about actual performance of entity detection systems on two manually annotated FgER datasets.

### 3.1 Is the Fine-ET type set an expansion of the extensively researched coarse-grained types?

For this analysis we manually inspected the most commonly used CgER dataset, CoNLL 2003. We analyzed how many entity types in the two popular Fine-ET hierarchies, FIGER and TypeNet are actual descendent of the four coarse-types present in the CoNLL dataset, namely *person*, *location*, *organization* and *miscellaneous*. The results are available in Figure 1. We can observe that in the FIGER typeset, 14% of types are not a descendants of the CoNLL types. This share increases in TypeNet where 25% of types are not descendants

| Models | FIGER | | | 1k-WFB-g | | |
|---|---|---|---|---|---|---|
| | Precision | Recall | F1 | Precision | Recall | F1 |
| LSTM-CNN-CRF (FIGER) | 87.17 | 28.95 | 43.47 | 91.41 | 37.13 | 52.81 |
| CoreNLP | 83.82 | 80.99 | 82.38 | 75.46 | 64.12 | 69.33 |
| NER Tagger | 80.44 | 84.01 | 82.19 | 77.25 | 68.52 | 72.62 |

Table 1: Performance of entity detection models trained on existing datasets evaluated on the FIGER and 1k-WFB-g datasets.

of CoNLL types. These types are from various diverse domain, including bio-medical, legal processes and entertainment and it is important in the aspect of the knowledge base construction applications to detect entity mentions of these types. These differences can be attributed to the fact that since 2003, the entity recognition problem has evolved a lot both in going towards finer-categorization as well as capturing entities from diverse domains.

### 3.2 How do entity detection systems perform in the Fine-ET setting?

For this analysis we evaluate two publicly available state-of-the-art entity detection systems, the Stanford CoreNLP [Manning et al., 2014] and the NER Tagger system proposed in [Lample et al., 2016]. Along with these, we also train a LSTM-CNN-CRF based sequence labeling model proposed in [Ma and Hovy, 2016] on the FIGER dataset. The learning models were evaluated on a manually annotated FIGER corpus and 1k-WFB-g corpus, a new in-house developed corpus specifically for FgER model evaluations. The results are presented in Table 1.

From the results, we can observe that a state-of-the-art sequence labeling model, LSTM-CNN-CRF trained on a dataset generated using NDS approach, such as FIGER dataset has lower recall compared with precision. On average the recall is 58% lower than precision. This is primarily because the NDS approach generates positive only annotations and the remaining un-annotated tokens contains large number of entity mentions. Thus the resulting dataset has large false negatives.

On the other hand, learning models trained on CoNLL dataset (CoreNLP and NER Tagger), have a much more balanced performance in precision and recall. This is because, being a manually annotated dataset, it is less likely that any entity mention (according to the annotation guidelines) will remain un-annotated. However, the recall is much lower (16% lower) on the 1k-WFB-g corpus as on the FIGER corpus. This is because, when designing 1k-WFB-g we insured that it has sufficient examples covering 117 entity types. Whereas, the FIGER evaluation corpus has only has 42 types of entity mentions and 80% of mentions are from *person*, *location* and *organization* coarse types. These results also highlight the coverage issue, mentioned in section 3.1. When the evaluation set is balanced covering a large spectrum of entity types, the performance of models trained on the CoNLL dataset goes down because of presence out-of-scope entity types. An ideal entity detection system should be able to work on the traditional as well as other entities relevant to FgER problem, i.e., good performance across all types. A statistical comparison of FIGER and 1k-WFB-g corpus is provided in Table 2.

The use of CoreNLP or learning models trained on CoNLL dataset is a standard practice to detect entity mentions in existing FgER research [Ling and Weld, 2012]. Our analysis

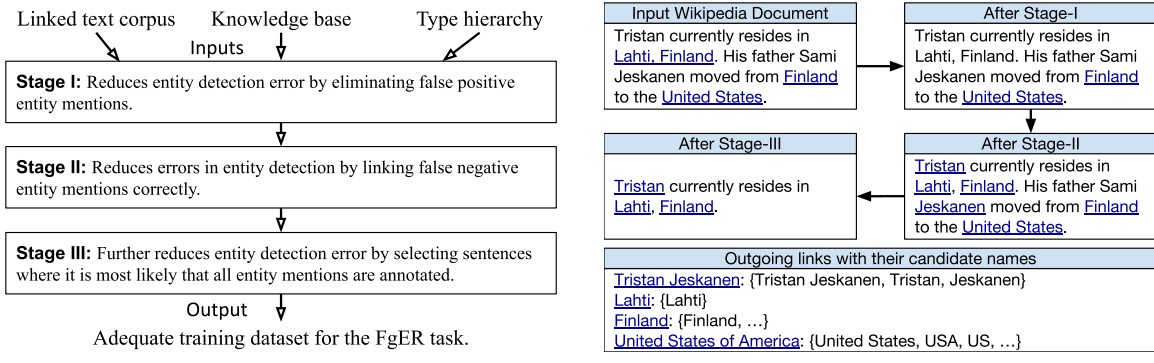

(a) A high-level overview of the three stages of HAnDS framework along with each stage's objective.

(b) The top four boxes illustrate how annotations changes during different stages. The bottom box illustrates the outgoing links and the candidate names for the example document.

Figure 2: An overview of HAnDS framework (left) along with an illustration on the framework in action on an example document (right).

conveys that this practice has its limitation in terms of detecting entities which are out of the scope of the CoNLL dataset. In the next section, we will describe our approach of automatically creating a training dataset for the FgER task. The same learning models, when trained on our generated training datasets will have a better and a balanced precision and recall.

## 4. HAnDS Framework

The objective of the HAnDS framework is to automatically create a corpus of sentences where every entity mention is correctly detected and is being characterized into one or more entity types. The scope of entities, i.e., what types of entities should be annotated is decided by a type hierarchy, which is one of the inputs of the framework. Figure 2 gives an overview of the HAnDS framework.

### 4.1 Inputs

The framework requires three inputs, a linked text corpus, a knowledge base and a type hierarchy.

**Linked text corpus:** A linked text corpus is a collection of documents where sporadically important concepts are hyperlinked to another document. For example, Wikipedia is a large-scale multi-lingual linked text corpus. The framework considers the span of hyperlinked text (or anchor text) as potential candidates for entity mentions.

**Knowledge base:** A knowledge base (KB) captures concepts, their properties, and inter-concept properties. Freebase, WikiData [Vrandečić and Krötzsch, 2014] and UMLS [Bodenreider, 2004] are examples of popular knowledge bases. A KB usually has a type property where multiple fine-grained semantic types/labels are assigned to each concept.

**Type hierarchy:** A type hierarchy ($\mathcal{T}$) is a hierarchical organization of various entity

types. For example, an entity type *city* is a descendant of type *geopolitical entity*. There have been various hierarchical organization schemes of fine-grained entity types proposed in literature, which includes, a 200 type scheme proposed in [Sekine, 2008], a 113 type scheme proposed in [Ling and Weld, 2012], a 87 type scheme proposed in [Gillick et al., 2014] and a 1081 type scheme proposed in [Murty et al., 2018]. However, in our work, we use two such hierarchies, FIGER[2] and TypeNet. FIGER being the most extensively used hierarchy and TypeNet being the latest and largest entity type hierarchy.

## 4.2 The three stages of the HAnDS framework

Automatic corpora creation using distant supervised methods inevitably will contain errors. For example, in the context of FgER, the errors could be at annotating entity boundaries, i.e, entity detection errors, or assigning an incorrect type, i.e., entity linking errors or both. The three-step process in our proposed HAnDS framework tries to reduce these errors.

### 4.2.1 Stage-I: Link categorization and Preprocessing

The objective of this stage is to reduce false positives entity mentions, where an incorrect anchor text is detected as an entity mention. To do so, we first categorize all hyperlinks of the document being processed as *entity links* and *non-entity links*. Further, every link is assigned a tag of being a *referential link* or not.

**Entity links:** These are a subset of links whose anchor text represents candidate entity mentions. If the labels obtained by a KB for a link, belongs to $\mathcal{T}$, we categorize that link as an entity link. Here, the $\mathcal{T}$ decides the scope of entities in the generated dataset. For example, if $T$ is the FIGER type hierarchy, then the hyperlink photovoltaic cell is not an entity link as its labels obtained by Freebase is not present in $T$. However, if $T$ is the TypeNet hierarchy, then photovoltaic cell is an entity link of type *invention*.

**Non-entity links:** These are a subset of links whose anchor text does not represent an entity mention. Since knowledge bases are incomplete, if a link is not categorized as an *entity link* it does not mean that the link will not represent an entity. We exploit corpus level context to categorize a link as a *non-entity link* using the following criteria: across complete corpus, the link should be mentioned at least 50 times (support threshold) and at least 50% of times (confidence threshold) with a lowercase anchor text. The intuition of this criteria is that we want to be certain that a link actually represents a non-entity. For example, this heuristic categorizes RBI as a *non-entity link* as there is no label present for this link in Freebase. Here RBI refers to the term "run batted in", frequently used in the context of baseball and softball. Unlike, [Nothman et al., 2008] which discards non entity mentions to have capitalized word, our data-driven heuristics does not put any hard constraints.

**Referential links:** A link is said to be referential if its anchor text has a direct case-insensitive match with the list of allowed candidate names for the linked concept. A KB can provide such list. For example, for an entity `Bill Gates`, the candidate names provided by Freebase includes `Gates` and `William Henry Gates`. However, in Wikipedia, there exists

---

2. Based on our observations, we made a few changes to the original FIGER hierarchy (seven additions, one correction, one merger, one deletion, and one substitute).

hyperlinks such as Bill and Melinda Gates linking to Bill Gates page, which is erroneous as the hyperlinked text is not the correct referent of the entity `Bill Gates`.

After categorization of links, except for *referential entity links*, we unlink all other links. Unlinking non-referential links such as Bill and Melinda Gates reduce *entity detection errors* by eliminating false positive entity mentions. The unlinked text span or a part of it can be referential mention for some other entities, as in the above example `Bill` and `Melinda Gates`. Figure 2b also illustrates this process where Lahti, Finland get unlinked after this stage. The next stage tries to re-link the unlinked tokens correctly.

### 4.2.2 STAGE-II: INFER ADDITIONAL LINKS

The objective of this stage is to reduce false negative entity mentions, where an entity mention is not annotated. This is done by linking the correct referential name of the entity mention to the correct node in KB.

To reduce entity linking errors, we use the document level context by restricting the candidate links (entities or non-entities) to the outgoing links of the current document being processed. For example, in Figure 2b, while processing an article about an Finnish-American luger Tristan Jeskanen, it is unlikely to observe mention of a 1903 German novel having the same name, i.e., Tristan.

To reduce false negative entity mentions, we construct two trie trees capturing the outgoing links and their candidates referential names for each document. The first trie contains all links and the second trie only contains links of entities which are predominantly expressed in lowercase phrases[3] (e.g. names of diseases). For each non-linked uppercase character, we match the longest matching prefix string within the first trie and assign the matching link. In the remaining non-linked phrases, we match the longest matching prefix string within the second trie and assign the matching link. Linking the candidate entities in unlinked phrases reduces entity detection error, by eliminating false negative entity mentions.

Unlike [Nothman et al., 2008], the two step string matching process ensures the possibility of a lowercase phrase being an entity mention (e.g. *lactic acid*, *apple juice*, *bronchoconstriction*, etc.) and a word with a first uppercase character being a non-entity (e.g. *Jazz*, *RBI*,[4] etc.).

Figure 2b shows an example of the input and output of this stage. In this stage, the phrases *Tristan*, *Lahti*, *Finland* and *Jeskanen* gets linked.

### 4.2.3 STAGE-III: SENTENCE SELECTION

The objective of this stage is to further reduce entity detection errors. This stage is motivated by the incomplete nature of practical knowledge bases. KBs do not capture all entities present in a linked text corpus and do not provide all the referential names for an entity mention. Thus, after stage-II there will be still a possibility of having both types of entity detection errors, false positives, and false negatives.

To reduce such errors in the induced corpus, we select sentences where it is most likely that all entity mention are annotated correctly. The resultant corpora of selected sentences

---

3. More than 50% of anchor text across corpus should be a lowercase phrase.
4. A run batted in (RBI) is a statistic in baseball and softball.

| Data sets | Wiki-FbF | Wiki-FbT | 1k-WFB-g | FIGER (GOLD) |
|---|---|---|---|---|
| # of sentences | $31,896,989$ | $32,583,731$ | 982 | 434 |
| # of entities | $37,734,727$ | $45,697,034$ | $2,420$ | 563 |
| # of unique entities | $2,506,518$ | $2,557,122$ | - | - |
| # of unique mentions | $3,264,876$ | $3,427,161$ | $2,151$ | 331 |
| # of tokens | $690,086,692$ | $707,347,974$ | $25,658$ | $10,008$ |
| # of unique tokens | $2,250,565$ | $2,280,446$ | $7,245$ | $2,578$ |
| $\mu$ sentence length | 21.63 | 21.71 | 26.13 | 23.06 |
| $\mu$ label per entity | 2.12 | 9.60 | 1.64 | 1.38 |
| # of types | 118 | 1115 | 117 | 43 |

Table 2: Statistics of the different datasets generated or used in this work.

will be our final dataset. To select these sentences, we exploit sentence-level context by using POS tags and list of the frequent sentence starting words. We only select sentences where all unlinked tokens are most likely to be a non-entity mention. If an unlinked token has a capitalized characters, then it likely to be an entity mention. We do not select such sentences, except in the following cases. In the first case, the token is a sentence starter, and is either in a list of frequent sentence starter word[5] or its POS tag is among the list of permissible tags[6]. In the second case, the token is an adjective, or belongs to occupational titles or is a name of day or month.

Figure 2b shows an example of the input and output of this stage. Here only the first sentence of the document is selected because in the other sentence the name **Sami** is not linked. The sentence selection stage ensures that the selected sentences have high-quality annotations. We observe that only around 40% of sentences are selected by stage III in our experimental setup.[7] Our extrinsic analysis in Section 5.2 shows that this stage helps models to have a significantly better recall.

In the next section, we describe the dataset generated using the HAnDS framework along with its evaluations.

## 5. Dataset Evaluation

Using the HAnDS framework we generated two datasets as described below:
**WikiFbF:** A dataset generated using Wikipedia, Freebase and the FIGER hierarchy as an input for the HAnDS framework. This dataset contains around 38 million entity mentions annotated with 118 different types.
**WikiFbT:** A dataset generated using Wikipedia, Freebase and the TypeNet hierarchy as an input for the HAnDS framework. This dataset contains around 46 million entity mentions annotated with 1115 different types.

In our experiments, we use the September 2016 Wikipedia dump. Table 2 lists various statistics of these datasets. In the next subsections, we estimate the quality of the generated datasets, both intrinsically and extrinsically. Our intrinsic evaluation is focused on

---

5. 150 most frequent words were used in the list.

6. POS tags such as DT, IN, PRP, CC, WDT etc. that are least likely to be candidate for entity mention.

7. An analysis of several characteristics of the discarded and retained sentences in available in the supplementary material at: https://github.com/abhipec/HAnDS.

| | WikiFbT | WikiFbF | | WikiFbT | WikiFbF |
|---|---|---|---|---|---|
| $\mathcal{H}$ | $45,696,943$ | $37,734,658$ | $\mathcal{H}$ | $2,557,122$ | $2,506,518$ |
| $\mathcal{N}$ | $24,594,804$ | $20,590,776$ | $\mathcal{N}$ | $1,630,078$ | $1,585,518$ |
| $\mathcal{H}-\mathcal{N}$ | $22,585,152$ | $18,261,738$ | $\mathcal{H}-\mathcal{N}$ | $959,694$ | $952,638$ |
| $\mathcal{H}\cap\mathcal{N}$ | $23,111,791$ | $19,472,920$ | $\mathcal{H}\cap\mathcal{N}$ | $1,597,428$ | $1,553,880$ |
| $\mathcal{N}-\mathcal{H}$ | $1,483,013$ | $1,117,856$ | $\mathcal{N}-\mathcal{H}$ | $32,650$ | $31,638$ |

| (a) Analysis of entity mentions. | (b) Analysis of entities. |
|---|---|

Table 3: Quantitative analysis of dataset generated using the HAnDS framework with the NDS approach of dataset generation. Here $\mathcal{H}$ denotes a set of entity mentions in Table 3a and set of entities in Table 3b generated by the HAnDS framework, and $\mathcal{N}$ denotes a set of entity mentions in Table 3a and set of entities in Table 3b generated by the NDS approach.

quantitative analysis, and the extrinsic evaluation is used as a proxy to estimate precision and recall of annotations.

## 5.1 Intrinsic evaluation

In intrinsic evaluation, we perform a quantitative analysis of the annotations generated by the HAnDS framework with the NDS approach. The result of this analysis is presented in Table 3. We can observe that on the same sentences, HAnDS framework is able to generate about 1.9 times more entity mention annotations and about 1.6 times more entities for the WikiFbT corpus compared with the NDS approach. Similarly, there are around 1.8 times more entity mentions and about 1.6 time more entities in the WikiFbF corpus. In Section 5.2.4, we will observe that despite around 1.6 to 1.9 times more new annotations, these annotations have a very high linking precision. Also, there is a large overlap among annotations generated using HAnDS framework and NDS approach. Around above 95% of entity mentions (and entities) annotations generated using the NDS approach are present in the HAnDS framework induced corpora. This indicated that the existing links present in Wikipedia are of high quality. The remaining 5% links were removed by the HAnDS framework as false positive entity mentions.

## 5.2 Extrinsic evaluation

In extrinsic evaluation, we evaluate the performance of learning models when trained on datasets generated using the HAnDS framework. Due to resource constraints, we perform this evaluation only for the WikiFbF dataset and its variants.

### 5.2.1 Learning Models

Following [Ling and Weld, 2012] we divided the FgER task into two subtasks: Fine-ED, a sequence labeling problem and Fine-ET, a multi-label classification problem. We use the existing state-of-the-art models for the respective sub-tasks. The FgER model is a simple pipeline combination of a Fine-ED model followed by a Fine-ET model.

**Fine-ED model:** For Fine-ED task we use a state-of-the-art sequence labeling based LSTM-CNN-CRF model as proposed in [Ma and Hovy, 2016].

**Fine-ET model:** For Fine-ET task we use a state-of-the-art LSTM based model as proposed in [Abhishek et al., 2017].

Please refer to the respective papers for model details.[8] The values of various hyperparameters used in the models along with the training procedure is mentioned in the supplementary material available at: https://github.com/abhipec/HAnDS.

### 5.2.2 Datasets

The two learning models are trained on the following datasets:

(1) **Wiki-FbF:** Dataset created by the HAnDS framework.

(2) **Wiki-FbF-w/o-III:** Dataset created by the HAnDS framework without using stage III of the pipeline.

(3) **Wiki-NDS:** Dataset created using the naive distant supervision approach with the same Wikipedia version used in our work..

(4) **FIGER:** Dataset created using the NDS approach but shared by [Ling and Weld, 2012].

Except for the FIGER dataset, for other datasets, we randomly sampled two million sentences for model training due to computational constraints. However, during model training as described in the supplementary material, we ensured that every model irrespective of the dataset, is trained for approximately same number of examples to reduce any bias introduced due to difference in the number of entity mentions present in each dataset. All extrinsic evaluation experiments, subsequently reported in this section are performed on these randomly sampled datasets. Also, the same dataset is used to train Fine-ED and Fine-ET learning model. This setting is different from [Ling and Weld, 2012] where entity detection model is trained on the CoNLL dataset. Hence, the result reported in their work is not directly comparable.

We evaluated the learning models on the following two datasets:

(1) **FIGER:** This is a manually annotated evaluation corpus which has been created by [Ling and Weld, 2012]. This contains 563 entity mentions and overall 43 different entity types. The type distribution in this corpus is skewed as only 11 entity types are mentioned more than 10 times.

(2) **1k-WFB-g:** This is a new manually annotated evaluation corpus developed specifically to cover large type set. This contains 2420 entity mentions and overall 117 different entity types. In this corpus 84 entity types are mentioned more than 10 types. The sentences for this dataset construction were sampled from Wikipedia text.

The statistics of these datasets is available in Table 2.

### 5.2.3 Evaluation Metric

For the Fine-ED task, we evaluated model's performance using the *precision*, *recall* and *F1* metrics as computed by the standard conll evaluation script[9]. For the Fine-ET and

---

8. Please note that there are several other models with competitive or better performance such as [Chiu and Nichols, 2016, Lample et al., 2016, Ma and Hovy, 2016] for sequence labeling problem and [Ren et al., 2016a, Shimaoka et al., 2017, Abhishek et al., 2017, Xin et al., 2018, Xu and Barbosa, 2018] for multi-label classification problem. Our criteria for model selection was simple; easy to use publicly available efficient implementation.

9. https://www.clips.uantwerpen.be/conll2002/ner/bin/conlleval.txt

| Models | FIGER | | | 1k-WFB-g | | |
|---|---|---|---|---|---|---|
| | Precision | Recall | F1 | Precision | Recall | F1 |
| LSTM-CNN-CRF (FIGER) | **87.17** | 28.95 | 43.47 | 91.41 | 37.13 | 52.81 |
| CoreNLP | 83.82 | 80.99 | 82.38 | 75.46 | 64.12 | 69.33 |
| NER Tagger | 80.44 | 84.01 | 82.19 | 77.25 | 68.52 | 72.62 |
| LSTM-CNN-CRF (Wiki-NDS) | 86.14 | 30.91 | 45.49 | **92.80** | 47.09 | 62.48 |
| LSTM-CNN-CRF (Wiki-FbF-w/o-III) | 88.07 | 44.58 | 59.2 | 92.55 | 65.03 | 76.39 |
| LSTM-CNN-CRF (Wiki-FbF) | 79.80 | **86.32** | **82.94** | 89.89 | **81.98** | **85.75** |

Table 4: Performance of the entity detection models on the FIGER and 1k-WFB-g datasets.

the FgER task, we use the *strict*, *loose-macro-average* and *loose-micro-average* evaluation metrics described in [Ling and Weld, 2012].

### 5.2.4 Result analysis for the Fine-ED task

The results of the entity detection models on the two evaluation datasets are presented in Table 4. From these results we perform two analysis. First, the effect of training datasets on model's performance and second, the performance comparison among the two manually annotated datasets.

In the first analysis, we observe that the LSTM-CNN-CRF model when trained on Wiki-FbF dataset has the highest F1 score on both the evaluation corpus. Moreover, the average difference in precision and recall for this model is the lowest, which indicates a balanced performance across both evaluation corpus. When compared with the models trained on the NDS generated datasets (Wiki-NDS and FIGER), we observe that these models have best precision across both corpus, however, lowest recall. The result indicates that large number of false negatives entity mentions are present in the NDS induced datasets. In the case of model trained on the dataset Wiki-FbF-w/o-III dataset the performance is in between the performance of model trained on Wiki-NDS and Wiki-FbF datasets. However, they have a significantly lower recall on average around 28% lower than model trained on Wiki-FbF. This highlights the role of stage-III, by selecting only quality annotated sentence, erroneous annotations are removed, resulting in learning models trained on WikiFbF to have a better and a balanced performance.

In the second analysis, we observe that models trained on datasets generated using Wikipedia as sentence source, performs better on the 1k-WFB-g evaluation corpus as compared to the FIGER evaluation corpus. These datasets are FIGER training corpus, WikiFbF, Wiki-NDS and Wiki-FbF-w/o-III. The primarily reason for better performance is that the sentences constituting the 1k-WFB-g dataset were sampled from Wikipedia.[10] Thus, this evaluation is a same domain evaluation. On the other hand, FIGER evaluation corpus is based on sentences sampled from news and specialized magazines (photography and veterinary domains). It has been observed in the literature that in a cross domain evaluation setting, learning model performance is reduced compared to the same domain evaluation [Nothman et al., 2009]. Moreover, this result also conveys that to some extent learning

---

10. We ensured that the test sentences are not present in any of the training datasets.

| Training Datasets | FIGER | | | 1k-WFB-g | | |
|---|---|---|---|---|---|---|
| | Strict | Ma-F1 | Mi-F1 | Strict | Ma-F1 | Mi-F1 |
| FIGER | 25.07 | 34.56 | 36.47 | 27.76 | 35.14 | 37.31 |
| Wiki-NDS | 30.07 | 37.89 | 38.55 | 39.12 | 49.28 | 51.60 |
| Wiki-FbF | **56.31** | **70.70** | **68.23** | **53.34** | **68.42** | **69.23** |

Table 5: Performance comparison for the FgER task.

model trained on the large Wikipedia text corpus is also able to generalize on evaluation dataset consisting of sentences from news and specialized magazines.

Our analysis in this section as well as in Section 3.1 indicates that although the type coverage of FIGER evaluation corpus is low (43 types), it helps to better measure model's generalizability in a cross-domain evaluation. Whereas, 1k-WFB-g helps to measure performance across a large spectrum of entity types (117 types). Learning models trained on Wiki-FbF perform best on both of the evaluation corpora. This warrants the usability of the generated corpus as well as the framework used to generate the corpus.

### 5.2.5 Result analysis for the Fine-ET and the FgER task

We observe that for the Fine-ET task, there is not a significant difference between the performance of learning models trained on the Wiki-NDS dataset and models trained on the Wiki-FbF dataset. The later model performs approx 1% better in the micro-F1 metric computed on the 1k-WFB-g corpus. This indicates that in the HAnDS framework stage-II, where false negative entity mentions were reduced by relinking them to Freebase, has a very high linking precision similar to NDS, which is estimated to be about 97-98% [Wang et al., 2016].

The results for the complete FgER system, i.e., Fine-ED followed by Fine-ET are available in Table 5. These results supports our claim in Section 3.1, that the current bottleneck for the FgER task, is Fine-ED, specifically lack of resource with quality entity boundary annotations while covering large spectrum of entity types. Our work directly addressed this issue. In the FgER task performance measure, learning model trained on WikiFbF has an average absolute performance improvement of at least 18% on all of the there evaluation metrics.

## 6. Conclusion and Discussion

In this work, we initiate a push towards moving from CgER systems to FgER systems, i.e., from recognizing entities from a handful of types to thousands of types. We propose the HAnDS framework to automatically construct quality training dataset for different variants of FgER tasks. The two datasets constructed in our work along with the evaluation resource are currently the largest available training and testing dataset for the entity recognition problem. They are backed with empirical experimentation to warrants the quality of the constructed corpora.

The datasets generated in our work opens up two new research directions related to the entity recognition problem. The first direction is towards an exploration of sequence labeling approaches in the setting of FgER, where each entity mention can have more than

one type. The existing state-of-the-art sequence labeling models for the CgER task, can not be directly applied in the FgER setting due to state space explosion in the multi-label setting. The second direction is towards noise robust sequence labeling models, where some of the entity boundaries are incorrect. For example, in our induced datasets, there are still entity detection errors, which are inevitable in any heuristic approach. There has been some work explored in [Dredze et al., 2009] assuming that it is a priori known which tokens have noise. This information is not available in our generated datasets.

Additionally, the generated datasets are much richer in entity types compared to any existing entity recognition datasets. For example, the generated dataset contains entities from several domains such as biomedical, finance, sports, products and entertainment. In several downstream applications where NER is used on a text writing style different from Wikipedia, the generated dataset is a good candidate as a source dataset for transfer learning to improve domain-specific performance.

## Acknowledgments

We would like to thank the anonymous reviewers for their insightful comments. We also thank Nitin Nair for his help with code to convert data from brat annotation tool to different formats. We acknowledge the use of computing resources made available from the Board of Research in Nuclear Science (BRNS), Dept. of Atomic Energy (DAE), Govt. of India sponsored project (No.2013/13/8-BRNS/10026) by Dr. Aryabartta Sahu at Department of Computer Science and Engineering, IIT Guwahati. Abhishek is supported by MHRD fellowship, Government of India.

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
