# OpenReview forum: "Fine-grained Entity Recognition with Reduced False Negatives and Large Type Coverage"
_AKBC.ws/2019/Conference — AKBC 2019_

### Official Review · AnonReviewer2 · 2018-12-21
**Good paper with two caveats**

**Rating:** 7
**Confidence:** 4

**Review:**

This paper presents a methodology for automatically annotating texts from Wikipedia with entity spans and types. The intended use is the generation of training sets for NER systems with a type hierarchy going beyond the common four types per, loc, org, misc.

I think this paper is a good fit for AKBC, given that entity recognition and linking systems are an essential part of knowledge-base population. This should be good for interesting discussions.

A big caveat of this paper is that, while the approach discussed by the authors is generally sound, it is very tailored towards Wikipedia texts, and it is unclear how it can be generalized beyond. Since this approach is meant for downstream training of NER systems, these systems will be limited to Wikipedia-style texts. A statement from the authors regarding this limitation would have been nice. Or maybe it's not a limitation at all? If the authors could show that NER systems trained on this Wikipedia-style texts do perform well on, say, scientific texts, that would be already good.

A second issue I see is that Stage III of the algorithm filters out sentences that might have missing entity labels. This makes sense, provided that the discarded sentences are not fundamentally different from the sentences retained in the NER tagger training set. For example, they could be longer on average, or could have more subordinate clauses, or just more entities on average, etc. This is something the authors should look into and should report in their paper. If they find that there are differences, an experiment would be nice that applies an NER tagger trained on the produced data to these sentences and verifies its performance there.

A minor point: The authors claim their approach scales to thousands of entity types, which I find a bit of an overstatement, given that the dataset produced by the authors has <1.5k different types (which is already quite a lot).

---

> ### Author Response · Authors · 2019-01-24
> **Thanks for the review. Indeed the issues highlighted are important and we hope that our response will address these issues.**
>
>
> Comment 1:
> The approach is tailored towards Wikipedia text.
>
> Response to comment 1:
> We agree that all our experiments use Wikipedia as a text source. In fact, to the best of our knowledge, only Wikipedia text exists in public domain that is highly interlinked to knowledge bases such as Freebase, DBpedia and WikiData. Our approach fundamentally requires a linked text corpus, and the current options are limited to Wikipedia only.
>
> Also, we would like to mention here is that FIGER evaluation dataset consists of sentences sampled from news and specialized magazines (photography and veterinary domain). Thus, it has a different writing style as compared with Wikipedia text. In Table 4, we can observe that the entity detection model trained on our dataset outperforms models trained on other existing datasets. Hence, the learning model is able to generalize beyond the Wikipedia style text.
>
> Further, for several other domains, such as scientific articles, conversation text, where directly utilizing the generated datasets might not yield good performance, we consider our datasets as a source dataset for transfer learning. The generated dataset is richer in entity types (1k+ from several domains) and is very large, which makes it a good candidate to use as a source dataset for transfer learning setting when the task is to do NER for a specific domain text.
>
> We have updated the discussion section (last paragraph) and 5.2.4 section (3rd paragraph, last sentence) to reflect this view.
>
>
> Comment/Question 2:
> Are discarded and retained sentences fundamentally same or different?
>
> Response to comment/question 2:
> This is an interesting question. To answer the question we analyzed the discarded and retained sentences on the following parameters:
> 1. Are the discarded sentences longer on average?
> 2. Does the entity mention in discarded sentences longer on average?
> 3. Is there a fundamental change in the token and entity mention distribution of discarded and retained sentences?
>
> This analysis is added as Appendix B to the paper. The summary of the analysis is that, other than the sentence length distribution, there is not a significant change in other distributions. The mean sentence length in the retained corpus is around 22, whereas in the discarded corpus is around 27.
>
> Note that, this result has a subtle interpretation. We observe that in the discarded sentences, there are more than 100k sentences with the length greater than 100 tokens. In fact, a corpus constituting of only these longer sentences is larger than several news-domain NER datasets. We observe that the majority of these sentences are caused due to incorrect sentence segmentations or they follow a list like patterns such as:
> 1. PER, PER, PER, PER, PER …
> 2. PER - PER - PER - PER - PER …
> 3. Director (Movie), Director (Movie), Director (Movie), ....
> 4. Project (year), project (year), project (year), …
> 5. NUMBER NUMBER NUMBER NUMBER …
> 6. Movie (year), movie (year), movie (year), …
> 7. PER | PER | PER | PER | PER ...
>
> The largest sentence length in discarded sentences is 6564 tokens!!
>
> Our dataset also captures these long sentences but the number is far less: 7664 sentences with the length greater than 100. The largest sentence length in the retained sentences is 624 tokens.
>
> Although, being a basic analysis, the analysis conveys that these longer sentences might not be suitable for applications where NER systems are used. To support this claim, we plotted the sentence length distribution of five NER datasets from different domains in Figure 7 and 8. The result conveys that, sentences longer than 100 words occur rarely in these domains and the sentence length distribution in the retained sentences is closer to the sentence length distribution in these domains when compared with the discarded sentences.
>
>
> Comment 3:
> scales to thousands of types is a bit overstatement
>
> Response to comment 3:
> We agree with the reviewer that the current experiments demonstrate the efficacy of the framework for up to 1.1k types. We would like to add on this that the Freebase (which has 1.1k entity types) can be replaced with WikiData (which as 15k entity types) in the HAnDS framework to generate a training dataset constituting of 15k entity types. However, since we don’t have any empirical results on this, we will modify the scalability claim in the paper.
>
>
> We hope that our response will provide a better insight into the proposed work and will clarify your concerns.

---

### Official Review · AnonReviewer3 · 2019-01-10
**New distant supervision method focusing on mention coverage, but incomplete evaluation**

**Rating:** 4
**Confidence:** 4

**Review:**

The paper proposes a new way to get distant supervision for fine grained NER using wikipedia, freebase and a type system. The paper explores using links in wikipedia as entities, matching them to freebase, and then expanding the using some string matching, and then finally pruning sentences using heuristics. Methods are compared on FIGER. The paper also introduces a dataset, but it is not fully described.

One interesting aspect about this paper is, as far as I can tell, one of the few works actually doing mention detection, and exploring the role mention detection.

That being said, its unclear what is new about the proposed source of supervision. The first two stages seem similar to standard methods and the last method (generally speaking pruning noisy supervision), has also been explored (e.g. in the context of distant supervision for IE, http://www.aclweb.org/anthology/P18-2015, and see Section 2.2) . Its also not clear to me what specifically targets good mention detection in this method. The experiments do somewhat argue that mention detection is improving, but not really on FIGER, but instead on the new dataset (this inconsistency causes me some pause).

All that being said, I don't think it would matter much if the supervision were incorporated into an existing system (e.g. https://github.com/uwnlp/open_type or https://github.com/MurtyShikhar/Hierarchical-Typing) and demonstrated competitive results (I understand that these systems use gt mention, but any stand in would be sufficient).  Table 5 on the other hand has some results that show baselines that are significantly worse than the original FIGER paper (without gt mentions) and the proposed supervision beating it (on the positive side, beating the original FIGER results too, but not included it in the table, see Section 3 of http://xiaoling.github.io/pubs/ling-aaai12.pdf ). So in the end, I'm not convinced on the experimental side.

This paper could be significantly improved on the evaluation. Incomplete reference to previous work on FIGER and insufficient description of their new datasets are areas of concern. Cursory reference, instead of evaluation, on new fine grained datasets (like open-type) also seem like missing pieces of a broader story introducing a new form of supervision.

---

> ### Author Response · Authors · 2019-01-24
> **Thanks for the critical review. We were intrigued by your comments. Our response is as follows:  (1/2)**
>
>
> Comment 1:
> Incomplete evaluation and mention detection not improving on FIGER as the results reported for the baseline used in our work is different for the results reported in the original FIGER paper (Table 2 of http://xiaoling.github.io/pubs/ling-aaai12.pdf)
>
> Response to comment 1:
> This is indeed a very interesting observation. We would like to clarify here that the results reported in the FIGER paper and in the baseline of the proposed work are not comparable due to the following reasons:
> 1. The datasets used are different as the FIGER paper does not use the FIGER dataset for a part of their experiments. This might seems surprising and can be easily missed as it is mentioned in the footnote 4 of the FIGER paper, decoupled with the result table.
> 2. The learning models are different.
>
> We followed a simple and consistent evaluation strategy for Table 5, which we summarize below:
> 1. Entity Recognition can be decomposed into two sub-problems: Entity Detection and Entity Classification
> 2. Use current state-of-the-art learning model for both the sub-problems in a pipelined manner and report the result for the entity recognition task.
> 3. The dataset used for the sub-problems is the same, as mentioned in Table 5. There is other interpretation of results as that of the results reported in the FIGER paper.
>
> We apologize for not bringing the correct interpretation of the results in the FIGER paper beforehand, which could have avoided this confusion.
>
> We would like to add here that mention detection is improving on both the datasets: FIGER and 1k-WFB-g. Please refer to Table 4. The resulting analysis of entity detection is available in Section 5.2.4. Also, there is an analysis of entity detection in Section 3. The improvement for mention detection is 0.08% F1 on FIGER dataset and 15.21% F1 on 1k-WFB-g. Note that although the improvement is small on FIGER, this improvement is over a model trained on a manually annotated (complete noise-free) dataset which matches well with FIGER dataset as both have predominant mentions of person, location and organization entity types. When we compare with a distantly supervised dataset, we have an improvement of 37.11% on FIGER dataset.
>
> We hope that this clarification will resolve the concerns about incomplete evaluation. We have updated Section 3 (last paragraph) and Section 5.2.2 (last paragraph) to clearly state that the results in FIGER paper are not directly comparable.
>
> We will be glad to answer any other questions related to the evaluation.
>
>
> Comment 2:
> Resemblance to http://www.aclweb.org/anthology/P18-2015
>
> Response to comment 2:
> We agree that at an abstract level, the pruning of noisy dataset idea is the same. However, there are fundamental differences between these works:
> 1. The tasks are different, the referred paper by the reviewer is removing the false positives for relation extraction task and our work is removing false positives and false negatives for the entity detection task.
> 2. The fundamental approach for these two different tasks is also different, where the referred work is a modeling approach and our being a heuristic approach.
>
> It is possible to add a learning model to make certain decisions instead of a pure heuristic approach used in our framework. We consider this as a future work. This is also suggested by Reviewer 1.
>
> We would like to mention here that the main contributions of this work are:
> 1. An analysis that the existing coarsely annotated datasets such as CoNLL are not suitable to detect entity mentions in the context of FIGER and TypeNET type hierarchies. (Section 3)
> 2. A framework to automatically construct quality datasets suitable for fine entity detection and typing. (Section 4)
> 3. Establish the state-of-the-art baselines on FIGER and new manually annotated corpus which is much richer than FIGER evaluation corpus. (Section 5)
>
> Additionally, in the final version, we will update the related work section incorporating a reference to the work suggested by the reviewer along with other works which are also similar at an abstract level.
>
>
> Comment 3:
> Description of the new datasets is missing
>
> Response to comment 3:
> We would like to specify that the generated datasets description is mentioned in Section 5, overview paragraph. We also report various statistics of the datasets in Table 2. We have updated the Section 5.2.2 to include a description of evaluation datasets.
>
> We hope that our response will clarify your concerns.

---

> ### Author Response · Authors · 2019-01-24
> **Thanks for the critical review. We were intrigued by your comments. Our response is as follows: (2/2)**
>
>
> Comment 4:
> Supervision being used in existing entity typing systems: (TypeNet and Open-type)
>
> Response to comment 4:
> We would like to mention here that we proposed an approach to generate a training dataset for Fine-grained Entity Recognition (FgER) task. FgER task can be decomposed into two subproblems: Fine entity detection and fine entity typing. TypeNET and Open-type fall into the latter category. Our work has shown that as the number of types increase, entity detection task becomes the bottleneck for FgER task. Existing works such as FIGER, TypeNet and open-type do not address this issue. We established this point in Section 3 and proposed a framework for generating datasets to remove the bottleneck for FIGER and TypeNET hierarchy. Our benchmarking experiments were focused on providing quality estimates for the generated datasets. Thus we kept the learning model fixed and varied the dataset to demonstrate that the generated dataset help learning model to generalized better.
>
> Additionally, in the case of open-type, it is an elusive problem of detecting 10k+ types of entities as many of these types go beyond knowledge bases.
>
> We hope that our response will clarify your concerns.

---

### Official Review · AnonReviewer1 · 2019-01-11
**Super-exciting stuff, great analysis and overall a well written and executed paper**

**Rating:** 9
**Confidence:** 4

**Review:**

This paper studies the problem of fine grained entity recognition (\fger) and typing . They initially present an excellent analysis of how fine grained entity type sets are not a direct extension of the usual coarse grained types (person, organization, etc) and hence training \fger systems on data annotated on coarse grained typed datasets would lead to low recall systems. Secondly, they show that automatically created fine grained typed datasets are also not sufficient since it leads to low recall because of noisy distant supervision. This analysis was both an interesting read and also sets the stage for the main contribution of this work.

The main contribution of this work is to create a high recall large training corpora to facilitate \fger and typing. The authors propose a staged pipeline which takes in input (a) a text corpus linked to a knowledge base (using noisy distant supervision), (b) a knowledge base (and also a list of aliases for entities) and (c) a typed hierarchy. All of these are readily available. The output is a high-recall linked corpus with linked entities to the KB and the type hierarchy. To show the usability of the training corpus the paper performs excellent intrinsic and extrinsic analysis. In the intrinsic analysis, they show that the new corpus indeed has a lot more mentions annotated. In the extrinsic analysis, they show that state art of the models trained on the new corpus has very impressive gains in recall when compared to the original existing datasets and also wiki dataset with distant supervision. There is a loss in precision though, but it is fairly small when compared to the massive gains in recall. This experiments warrants the usability of the generated training corpus and also the framework they stipulate which I think everyone should follow.

Great work!

Questions:

1. In table 3, can you also report the difference in entities (and not entity mentions). I would be interested to see if you were able to find new entities.
2. Are you planning to release the two training datasets (and if possible the code ?)

Suggestion: The heuristics definitely work great but I think we can still do better if parts of stage  2 and 3 were replaced by a simple machine learned system. For example, in stage 2, just relying on heuristics to match to the prefix trees would result in always choosing the shortest alias and would be problematic if aliases of different entities share the same prefix. Restricting to entities in the document would definitely help here but still there might be unrecoverable errors. A simple model conditioned on the context would definitely perform better. Similarly in sentence 3, the POS based heuristics would just be more robust if a classifier is learned.

---

> ### Author Response · Authors · 2019-01-24
> **Thanks for an encouraging review! Here is our response to your questions:**
>
>
> Question 1:
> In Table 3, can you also report the differences in entities?
>
> Response to question 1:
> We have updated the Table 3 and section 5.1 to report differences in entities. The summary of the table is that our corpus has around 1.5 times more unique entities when compared with the NDS approach to annotate the same sentences. Also, there are slight changes due to a mistake in row 1 and 3 of WikiFbT numbers. The entity mentions annotated are now even more!  (42 million -> 45 million and 24 million -> 27 million).
>
> Question 2:
> Are you planning to release the two training datasets (and if possible the code ?)
>
> Response to question 2:
> All experiments reported in this work will be easily reproducible. A note for the same is included in the paper.
>
> We are planning to publicly release the following things:
> 1. The preprocessed Wikipedia text along with the preprocessed Freebase to facilitate the dataset construction research.
> 2. The code used to generate the two training datasets (WikiFbF and WikiFbT) along with the generated datasets.
> 3. The code used to train learning models and the evaluation corpus.
>
> Response to suggestion:
> Thanks for the great suggestion. Indeed there is a possibility of further improvement by replacing some of the heuristics with learning models. We consider this as a future work.

---

### Meta-Review · Area_Chair1 · 2019-02-12
**Good paper; but a heuristic approach**

**Recommendation:** Accept (Poster)
**Confidence:** 4

**Metareview:**

This paper design a framework based on heuristic approaches to automatically construct a dataset for the FgER task. The paper is solid and present nice experimental comparisons. Overall, it is clear and easy to follow.  However, as pointed out by the reviewers, the proposed approach is heuristic and may not be general enough for handling other tasks or data in other domains.

---

### Decision · Program_Chairs · 2019-02-15
**AKBC 2019 Conference Decision**

Accept